# Simple and Efficient Weighted Minwise Hashing

**Anshumali Shrivastava**
Department of Computer Science
Rice University
Houston, TX, 77005
anshumali@rice.edu

## Abstract

Weighted minwise hashing (WMH) is one of the fundamental subrou-tine, required by many celebrated approximation algorithms, commonly adopted in industrial practice for large -scale search and learning. The resource bottleneck with WMH is the computation of multiple (typically a few hundreds to thousands) independent hashes of the data. We propose a simple rejection type sampling scheme based on a carefully designed red-green map, where we show that the number of rejected sample has exactly the same distribution as weighted minwise sampling. The running time of our method, for many practical datasets, is an order of magnitude smaller than existing methods. Experimental evaluations, on real datasets, show that for computing 500 WMH, our proposal can be 60000x faster than the Ioffe's method without losing any accuracy. Our method is also around 100x faster than approximate heuristics capitalizing on the efficient "densified" one permutation hashing schemes [26, 27]. Given the simplicity of our approach and its significant advantages, we hope that it will replace existing implementations in practice.

## 1 Introduction

(Weighted) Minwise Hashing (or Sampling), [2, 4, 17] is the most popular and successful randomized hashing technique, commonly deployed in commercial big-data systems for reducing the computational requirements of many large-scale applications [3, 1, 25].

Minwise sampling is a known LSH for the Jaccard similarity [22]. Given two positive vectors $x, y \in \mathbb{R}^{\mathbb{D}}$, $x, y > 0$, the (generalized) Jaccard similarity is defined as

$$\mathbb{J}(x, y) = \frac{\sum_{i=1}^{D} \min\{x_i, y_i\}}{\sum_{i=1}^{D} \max\{x_i, y_i\}}. \tag{1}$$

$\mathbb{J}(x, y)$ is a frequently used measure for comparing web-documents [2], histograms (specially images [13]), gene sequences [23], etc. Recently, it was shown to be a very effective kernel for large-scale non-linear learning [15]. WMH leads to the best-known LSH for $L_1$ distance [13], commonly used in computer vision, improving over [7].

Weighted Minwise Hashing (WMH) (or Minwise Sampling) generates randomized hash (or fingerprint) $h(x)$, of the given data vector $x \geq 0$, such that for any pair of vectors $x$ and $y$, the probability of hash collision (or agreement of hash values) is given by,

$$Pr(h(x) = h(y)) = \frac{\sum \min\{x_i, y_i\}}{\sum \max\{x_i, y_i\}} = \mathbb{J}(x, y). \tag{2}$$

A notable special case is when $x$ and $y$ are binary (or sets), i.e. $x_i, y_i \in \{0, 1\}^D$. For this case, the similarity measure boils down to $\mathbb{J}(x, y) = \frac{\sum \min\{x_i, y_i\}}{\sum \max\{x_i, y_i\}} = \frac{|x \cap y|}{|x \cup y|}$.

Being able to generate a randomized signature, $h(x)$, satisfying Equation 2 is the key breakthrough behind some of the best-known approximations algorithms for metric labelling [14], metric embedding [5], mechanism design, and differential privacy [8].

A typical requirement for algorithms relying on minwise hashing is to generate, some large enough, $k$ independent Minwise hashes (or fingerprints) of the data vector $x$, i.e. compute $h_i(x)$ $i \in \{1, 2, ..., k\}$ repeatedly with independent randomization. These independent hashes can then be used for a variety of data mining tasks such as cheap similarity estimation, indexing for sublinear-search, kernel features for large scale learning, etc. The bottleneck step in all these applications is the costly computation of the multiple hashes, which requires multiple passes over the data. The number of required hashes typically ranges from few hundreds to several thousand [26]. For example, the number of hashes required by the famous LSH algorithm is $O(n^\rho)$ which grows with the size of the data. [15] showed the necessity of around 4000 hashes per data vector in large-scale learning with $\mathbb{J}(x, y)$ as the kernel, making hash generation the most costly step.

Owing to the significance of WMH and its impact in practice, there is a series of work over the last decade trying to reduce its costly computation cost [11].The first groundbreaking work on Minwise hashing [2] computed hashes $h(x)$ only for unweighted sets $x$ (or binary vectors), i.e. when the vector components $x_i$s can only take values 0 and 1. Later it was realized that vectors with positive integer weights, which are equivalent to weighted sets, can be reduced to unweighted set by replicating elements in proportion to their weights [10, 11]. This scheme was very expensive due to blowup in the number of elements caused by replications. Also, it cannot handle real weights. In [11], the authors showed few approximate solutions to reduce these replications.

Later [17], introduced the concept of consistent weighted sampling (CWS), which focuses on sampling directly from some well-tailored distribution to avoid any replication. This method, unlike previous ones, could handle real weights exactly. Going a step further, Ioffe [13] was able to compute the exact distribution of minwise sampling leading to a scheme with worst case $O(d)$, where $d$ is the number of non-zeros. This is the fastest known exact weighted minwise sampling scheme, which will also be our main baseline.

$O(dk)$ for computing $k$ independent hashes is very expensive for modern massive datasets, especially when $k$ with ranges up to thousands. Recently, there was a big success for the binary case, where using the novel idea of "Densification" [26, 27, 25] the computation time for unweighted minwise was brought down to $O(d + k)$. This resulted in over 100-1000 fold improvement. However, this speedup was limited only to binary vectors. Moreover, the samples were not completely independent.

Capitalizing on recent advances for fast unweighted minwise hashing, [11] exploited the old idea of replication to convert weighted sets into unweighted sets. To deal with non-integer weights, the method samples the coordinates with probabilities proportional to leftover weights. The overall process converts the weighted minwise sampling to an unweighted problem, however, at a cost of incurring some bias (see Algorithm 2). This scheme is faster than Ioffe's scheme but, unlike other prior works on CWS, it is not exact and leads to biased and correlated samples. Moreover, it requires strong and expensive independence [12].

All these lines of work lead to a natural question: does there exist an unbiased and independent WMH scheme with same property as Ioffe's hashes but significantly faster than all existing methodologies? We answer this question positively.

## 1.1 Our Contributions:

**1.** We provide an unbiased weighted minwise hashing scheme, where each hash computation takes time inversely proportional to effective sparsity (define later) which can be an order of magnitude (even more) smaller than $O(d)$. This improves upon the best-known scheme in the literature by Ioffe [13] for a wide range of datasets. Experimental evaluations on real datasets show more than 60000x speedup over the best known exact scheme and around 100x times faster than biased approximate schemes based on the recent idea of fast minwise hashing.
**2.** In practice, our hashing scheme requires much fewer bits usually (5-9) bits instead of 64 bits (or higher) required by existing schemes, leading to around 8x savings in space, as shown on real datasets.

**3.** We derive our scheme from elementary first principles. Our scheme is simple and it only requires access to uniform random number generator, instead of costly sampling and transformations needed by other methods. The hashing procedure is different from traditional schemes and could be of independent interest in itself. Our scheme naturally provide the quantification of when and how much savings we can obtain compared to existing methodologies.

**4.** Weighted Minwise sampling is a fundamental subroutine in many celebrated approximation algorithms. Some of the immediate consequences of our proposal are as follows:

- We obtain an algorithmic improvement, over the query time of LSH based algorithm, for $L_1$ distance and Jaccard Similarity search.
- We reduce the kernel feature [21] computation time with min-max kernels [15].
- We reduce the sketching time for fast estimation of a variety of measures, including $L_1$ and earth mover distance [14, 5].

## 2 Review: Ioffe's Algorithm and Fast Unweighted Minwise Hashing

We briefly review the state-of-the-art methodologies for Weighted Minwise Hashing (WMH). Since WMH is only defined for weighted sets, our vectors under consideration will always be positive, i.e. every $x_i \geq 0$. $D$ will denote the dimensionality of the data, and we will use $d$ to denote the number (or the average) of non-zeros of the vector(s) under consideration.

The fastest known scheme for exact weighted minwise hashing is based on an elegant derivation of the exact sampling process for "Consistent Weighted Sampling" (CWS) due to Ioffe [13], which is summarized in Algorithm 1. This scheme requires $O(d)$ computations.

$O(d)$ for a single hash computation is quite expensive. Even the unweighted case of minwise hashing had complexity $O(d)$ per hashes, until recently. [26, 27] showed a new one permutation based scheme for generating $k$ near-independent unweighted minwise hashes in $O(d + k)$ breaking the old $O(dk)$ barrier. However, this improvement does not directly extend to the weighted case. Nevertheless, it leads to a very powerful heuristic in practice.

---

**Algorithm 1** Ioffe's CWS [13]

**input** Vector $x$, random seed[][]

    **for** $i = 1$ **to** $k$ **do**
        **for** Iterate over $x_j$ s.t $x_j > 0$ **do**
            $randomseed = seed[i][j]$;
            Sample $r_{i,j}$, $c_{i,j} \sim Gamma(2,1)$.
            Sample $\beta_{i,j} \sim Uniform(0,1)$
            $t_j = \left\lfloor \frac{\log x_j}{r_{i,j}} + \beta_{i,j} \right\rfloor$
            $y_j = exp(r_{i,j}(t_j - \beta_{i,j}))$
            $z_j = y_j * exp(r_{i,j})$
            $a_j = c_{i,j}/z_j$
        **end for**
        $k^* = \arg\min_j a_j$
        $HashPairs[i] = (k^*, t_{k^*})$
    **end for**
    **RETURN** HashPairs[]

---

It was known that with some bias, weighted minwise sampling can be reduced to an unweighted minwise sampling using the idea of sampling weights in proportion to their probabilities [10, 14]. Algorithm 2 describes such a procedure. A reasonable idea is then to use the fast unweighted hashing scheme, on the top of this biased approximation [11, 24]. The inside for-loop in Algorithm 2 blows up the number of non-zeros in the returned unweighted set. This makes the process slower and dependent on the magnitude of weights. Moreover, unweighted sampling requires very costly random permutations for good accuracy [20].

---

**Algorithm 2** Reduce to Unweighted [11]

**input** Vector $x$,
    $S = \phi$
    **for** Iterate over $x_j$ s.t $x_j > 0$ **do**
        $floorx_j = \lfloor x_j \rfloor$
        **for** $i = 1$ **to** $floorx_j$ **do**
            $S = S \cup (i, j)$
        **end for**
        $r = Uniform(0,1)$
        **if** $r \leq x_j - floorx_j$ **then**
            $S = S \cup (floorx_j + 1, j)$
        **end if**
    **end for**
    **RETURN** S (unweighted set)

---

Both the Ioffe's scheme and the biased unweighted approximation scheme generate big hash values requiring 32-bits or higher storage per hash value. For reducing this to a manageable size of say 4-8 bits, a commonly adopted practical methodology is to randomly rehash it to smaller space at the cost of loss in accuracy [16]. It turns out that our hashing scheme generates 5-9 bits values, $h(x)$, satisfying Equation 2, without losing any accuracy, for many real datasets.

## 3 Our Proposal: New Hashing Scheme

We first describe our procedure in details. We will later talk about the correctness of the scheme. We will then discuss its runtime complexity and other practical issues.

### 3.1 Procedure

We will denote the $i^{th}$ component of vector $x \in \mathcal{R}^D$ by $x_i$. Let $m_i$ be the upper bound on the value of component $x_i$ in the given dataset. We can always assume the $m_i$ to be an integer, otherwise we take the ceiling $\lceil m_i \rceil$ as our upper bound. Define

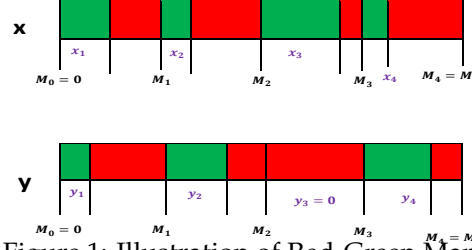

$$\sum_{k=1}^{i} m_i = M_i. \quad \text{and} \quad \sum_{k=1}^{D} m_i = M_D = M \quad (3)$$

If the data is normalized, then $m_i = 1$ and $M = D$.

Figure 1: Illustration of Red-Green Map of 4 dimensional vectors $x$ and $y$.

Given a vector $x$, we first create a red-green map associated with it, as shown in Figure 1. For this, we first take an interval $[0, M]$ and divide it into $D$ disjoint intervals, with $i^{th}$ interval being $[M_{i-1}, M_i]$ which is of the size $m_i$. Note that $\sum_{i=1}^{D} m_i = M$, so we can always do that. We then create two regions, red and green. For the $i^{th}$ interval $[M_{i-1}, M_i]$, we mark the subinterval $[M_{i-1}, M_{i-1} + x_i]$ as green and the rest $[M_{i-1} + x_i, M_i]$ with red, as shown in Figure 1. If $x_i = 0$ for some $i$, then the whole $i^{th}$ interval $[M_{i-1}, M_i]$ is marked as red.

Formally, for a given vector $x$, define the green $x_{green}$ and the red $x_{red}$ regions as follows

$$x_{green} = \cup_{i=1}^{D} [M_i, M_i + x_i]; \quad x_{red} = \cup_{i=1}^{D} [M_i + x_i, M_{i+1}]; \quad (4)$$

Our sampling procedure simply draws an independent random real number between $[0, M]$, if the random number lies in the red region we repeat and re-sample. We stop the process as soon as the generated random number lies in the green region. Our hash value for a given data vector, $h(x)$, is simply the number of steps taken before we stop. We summarize the procedure in Algorithm 3. More formally,

**Definition 1** *Define* $\{r_i : i = 1, 2, 3....\}$ *as a sequence of i.i.d uniformly generated random number between* $[0, M]$. *Then we define the hash of* $x$, $h(x)$ *as*

$$h(x) = \arg \min_i r_i, \quad s.t. \quad r_i \in x_{green} \quad (5)$$

Our procedure can be viewed as a form of rejection sampling [30]. To the best of our knowledge, there has been no prior evidence in the literature,

---

**Algorithm 3** Weighted MinHash

**input** Vector $x$, $M_i$'s, $k$, random seed[].

  Initialise Hashes[] to all 0s.
  **for** $i = 1$ **to** $k$ **do**
    $randomseed = seed[i]$;
    **while** true **do**
      $r = M \times Uniform(0, 1)$;
      **if** ISGREEN(r), (check if $r \in x_{red}$
      **then**
        break;
      **end if**
      $randomseed = \lceil r * 1000000 \rceil$;
      $Hashes[i] + +$;
    **end while**
  **end for**
  **RETURN** Hashes

---

where that the number of samples rejected has locality sensitive property.

We want our hashing scheme to be consistent [13] across different data points to guarantee Equation 2. This requires ensuring the consistency of the random numbers in hashes [13]. We can achieve the required consistency by pre-generating the sequence of random numbers and storing them analogous to other hashing schemes. However, there is an easy way to generate a fixed sequence of random numbers on the fly by ensuring the consistency of the random seed. This does not require any storage, except the starting seed. Our Algorithm 3 uses this criterion, to ensure the consistency of random numbers. We start with a fixed random seed for generating random numbers. If the generated random number lies in the red region, then before re-sampling, we reset the seed of our random number generator as a function of discarded random number. In the algorithm, we used $\lceil 100000 * r \rceil$, where $\lceil \rceil$ is the ceiling operation, as a convenient way to ensure the consistency of sequence, without any memory overhead. This seems to works nicely in practice. Since we are sampling real numbers, the probability of any repetition (or cycle) is zero. For generating $k$ independent hashes we just use different random seeds which are kept fixed for the entire dataset.

## 3.2 Correctness

We show that the simple, but very unusual, scheme given in Algorithms 3 actually does possess the required property, i.e. for any pair of points $x$ and $y$ Equation 2 holds. Unlike the previous works on this line [17, 13] which requires computing the exact distribution of associated quantities, the proof of our proposed scheme is elementary and can be derived from first principles. This is not surprising given the simplicity of our procedure.

**Theorem 1** *For any two vectors $x$ and $y$, we have*

$$Pr\left(h(x) = h(y)\right) = \mathbb{J}(x, y) = \frac{\sum_{i=1}^{D} \min\{x_i, y_i\}}{\sum_{i=1}^{D} \max\{x_i, y_i\}} \tag{6}$$

Theorem 1 implies that the sampling process is exact and we automatically have an unbiased estimator of $\mathbb{J}(x, y)$, using $k$ independently generated WMH, $h_i(x)$s from Algorithm 3.

$$\hat{J} = \frac{1}{k} \sum_{i=1}^{k} [\mathbf{1}\{h_i(x) = h_i(y)\}]; \quad \mathbb{E}(\hat{J}) = \mathbb{J}(x, y); \quad Var(\hat{J}) = \frac{\mathbb{J}(x, y)(1 - \mathbb{J}(x, y))}{k}, \tag{7}$$

where $\mathbf{1}$ is the indicator function.

## 3.3 Running Time Analysis and Fast Implementation
Define
$$s_x = \frac{\text{Size of green region}}{\text{Size of red region} + \text{Size of green region}} = \frac{\sum_{i=1}^{D} x_i}{M} = \frac{\|x\|_1}{M}, \tag{8}$$
as the ***effective sparsity*** of the vector $x$. Note that this is also the probability of $Pr(r \in x_{green})$. Algorithm 3 has a while loop.

We show that the expected times the while loops runs, which is also the expected value of $h(x)$, is the inverse of *effective sparsity* . Formally,

**Theorem 2**

$$\mathbb{E}(h(x)) = \frac{1}{s_x}; \quad Var(h(x)) = \frac{1 - s_x}{s_x^2}; \quad Pr\left(h(x) \geq \frac{\log \delta}{\log(1 - s_x)}\right) \leq \delta. \tag{9}$$

## 3.4 When is this advantageous over Ioffe's scheme?
The time to compute each hash value, in expectation, is the inverse of effective sparsity $\frac{1}{s}$. This is a very different quantity compared to existing solutions which needs $O(d)$. For datasets with $\frac{1}{s} \ll d$, we can expect our method to be much faster. For real datasets, such as image histograms, where minwise sampling is popular[13], the value of this sparsity is of the order of 0.02-0.08 (see Section 4.2) leading to $\frac{1}{s_x} \approx 13 - 50$. On the other hand, the number of non-zeros is around half million. Therefore, we can expect significant speed-ups.

**Corollary 1** *The expected amount of bits required to represent $h(x)$ is small, in particular,*

$$\mathbb{E}(bits) \leq -\log s_x; \quad \mathbb{E}(bits) \approx \log \frac{1}{s_x} - \frac{(1 - s_x)}{2}; \tag{10}$$

Existing hashing scheme require 64 bits, which is quite expensive. A popular approach for reducing space uses least significant bits of hashes [16, 13]. This tradeoff in space comes at the cost of accuracy [16]. Our hashing scheme naturally requires only few bits, typically 5-9 (see Section 4.2), eliminating the need for trading accuracy for manageable space.

We know from Theorem 2 that each hash function computation requires $\frac{1}{s}$ number of function calls to ISGREEN(r). If we can implement ISGREEN(r) in constant time, i.e $O(1)$, then we can generate generate $k$ independent hashes in total $O(d + \frac{k}{s})$ time instead of $O(dk)$ required by [13]. Note that $O(d)$ is the time to read the input vector which cannot be avoided. Once the data is loaded into the memory, our procedure is actually $O(\frac{k}{s})$ for computing $k$ hashes, for all $k \geq 1$. This can be a huge improvement as in many real scenarios $\frac{1}{s} \ll d$

Before we jump into a constant time implementation of ISGREEN(r), we would like readers to note that there is a straightforward binary search algorithm for ISGREEN(r) in $\log d$ time. We consider $d$ intervals $[M_i, M_i + x_i]$ for all $i$, such that $x_i \neq 0$. Because of the nature of the problem, $M_{i-1} + x_{i-1} \leq M_i \ \forall i$. Therefore, these intervals are disjoint and sorted. Therefore, given a random number $r$, determining if $r \in \cup_{i=1}^{D}[M_i, M_i + x_i]$ only needs binary search over $d$ ranges. Thus, in expectation, we already have a scheme that generates $k$ independent hashes in total $O(d + \frac{k}{s} \log d)$ time improving over best known $O(dk)$ required by [13] for exact unbiased sampling, whenever $d \gg \frac{1}{s}$.

We show that with some algorithmic tricks and few more data structures, we can implement ISGREEN(r) in constant time $O(1)$. We need two global pre-computed hashmaps, *IntToComp* (Integer to Vector Component) and *CompToM* (Vector Component to M value). *IntToComp* is a hashmap that maps every integer between $[0, M]$ to the associated components, i.e., all integers between $[M_i, M_{i+1}]$ are mapped to $i$, because it is associated with $i^{th}$ component. *Comp-ToM* maps every component of vectors $i \in \{1, 2, 3, ..., D\}$ to its associated value $M_i$. The procedure for computing these hashmaps is straightforward and is summarized in Al-

---

**Algorithm 4** ComputeHashMaps (Once per dataset)

---

**input** $M_i$'s,
  index =0, CompToM[0] =0
  **for** $i = 0$ **to** $D - 1$ **do**
    **if** $i < D - 1$ **then**
      $CompToM[i + 1] = M_i + CompToM[i]$
    **end if**
    **for** $j = 0$ **to** $M_i - 1$ **do**
      $IntToComp[index] = i$
      index++
    **end for**
  **end for**
  **RETURN CompToM[] and IntToComp[]**

---

gorithm 4. It should be noted that these hash-maps computation is a one time pre-processing operation over the entire dataset having a negligible cost. $M_i$'s can be computed (estimated) while reading the data.

Using these two pre-computed hashmaps, the ISGREEN(r) methodology works as follows: We first compute the ceiling of $r$, i.e. $\lceil r \rceil$, then we find the component $i$ associated with $r$, i.e., $r \in [M_i, M_{i+1}]$, and the corresponding associated $M_i$ using hashmaps *IntToComp* and *CompToM*. Finally, we return true if $r \leq x_i + M_i$ otherwise we return false. The main observation is that since we ensure that all $M_i$'s are Integers, for any real number r, if $r \in [M_i, M_{i+1}]$ then the same holds for $\lceil r \rceil$, i.e., $\lceil r \rceil \in [M_i, M_{i+1}]$. Hence we can work

---

**Algorithm 5** ISGREEN(r)

---

**input** $r$, $x$, Hashmaps *IntToComp[]* and *CompToM[]* from Algorithm 4.

  $index = \lceil r \rceil$
  $i = IntToComp[index]$
  $M_i = CompToM[i]$
  **if** $r \leq M_i + x_i$ **then**
    **RETURN TRUE**
  **end if**
  **RETURN FALSE**

---

with hashmaps using $\lceil r \rceil$ as the key. The overall procedure is summarized in Algorithm 5.

Note that our overall procedure is much simpler compared to Algorithm 1. We only need to generate random numbers followed by a simple condition check using two hash lookups. Our analysis shows that we have to repeat this only for small number of times. Compare this with the scheme of Ioffe where for every non-zero component of a vector we need to sample two Gamma variables followed by computing several expensive transformations including exponentials. We next demonstrate the benefits of our approach in practice.

## 4 Experiments

In this section, we demonstrate that in real high-dimensional settings, our proposal provides significant speedup and requires less memory over existing methods. We also need to validate our theory that our scheme is unbiased and should be indistinguishable in accuracy with Ioffe's method.

**Baselines:** Ioffe's method is the fastest known exact method in the literature, so it serves as our natural baseline. We also compare our method with biased unweighted approximations (see Algorithm 2) which capitalizes on recent success in fast unweighted minwise hashing [26, 27], we call it Fast-WDOPH (for Fast Weighted Densified One Permutation Hashing). Fast-WDOPH needs very long permutation, which is expensive. For efficiency,

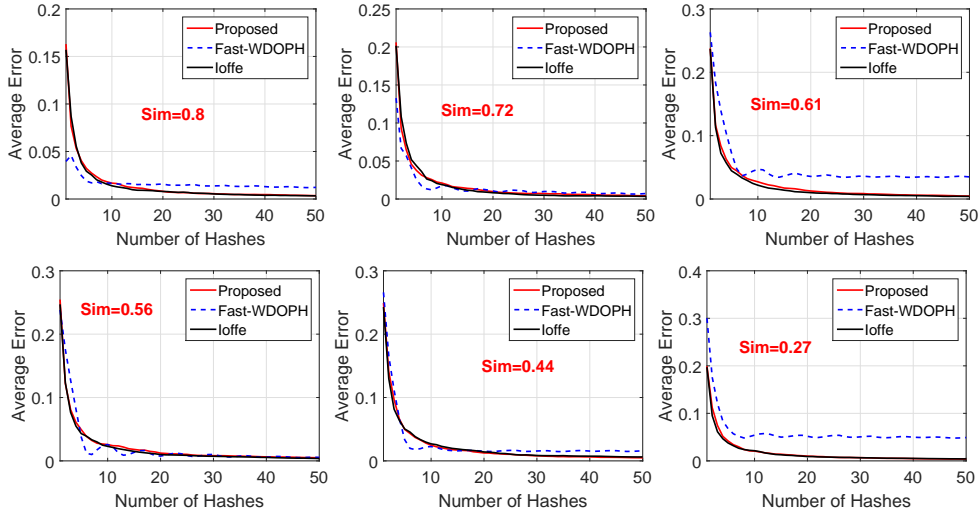

Figure 2: **Average Errors in Jaccard Similarity Estimation with the Number of Hash Values. Estimates are averaged over 200 repetitions.**

we implemented the permutation using fast 2-universal hashing which is always recommended [18].

**Datasets:** Weighted Minwise sampling is commonly used for sketching image histograms [13]. We chose two popular publicly available vision dataset **Caltech101** [9] and **Oxford** [19]. We used the standard publicly available Histogram

| Data | non-zeros (d) | Dim (D) | Sparsity (s) |
|---|---|---|---|
| Hist | 737 | 768 | 0.081 |
| Caltech101 | 95029 | 485640 | 0.024 |
| Oxford | 401879 | 580644 | 0.086 |

Table 1: Basic Statistics of the Datasets

of Oriented Gradient (HOG) codes [6], popular in vision task, to convert images into feature vectors. In addition, we also used random web images [29] and computed simple histograms of RGB values. We call this dataset as **Hist**. The statistics of these datasets is summarized in Table 1. These datasets cover a wide range of variations in terms of dimensionality, non-zeros and sparsity.

### 4.1 Comparing Estimation Accuracy

In this section, we perform a sanity check experiment and compare the estimation accuracy with WMH. For this task we take 9 pairs of vectors from our datasets with varying level of similarities. For each of the pair $(x, y)$, we generate $k$ weighted minwise hashes $h_i(x)$ and $h_i(y)$ for $i \in \{1, 2, .., k\}$, using the three competing schemes. We then compute the estimate of the Jac-

| Method | Prop | Ioffe | Fast-WDOPH |
|---|---|---|---|
| Hist | 10ms | 986ms | 57ms |
| Caltech101 | 57ms | 87105ms | 268ms |
| Oxford | 11ms | 746120ms | 959ms |

Table 2: Time taken in milliseconds (ms) to compute 500 hashes by different schemes. Our proposed scheme is significantly faster.

card similarity $\mathbb{J}(x, y)$ using the formula $\frac{1}{k} \sum_{i=1}^{k} [\mathbf{1}\{h_i(x) = h_i(y)\}]$ (See Equation 7). We compute the errors in the estimate as a function of $k$. To minimize the effect of randomization, we average the errors from 200 random repetitions with different seeds. We plot this average error with $k = \{1, 2, ..., 50\}$ in Figure 2 for different similarity levels.

We can clearly see from the plots that the accuracy of the proposed scheme is indistinguishable from Ioffe's scheme. This is not surprising because both the schemes are unbiased and have the same theoretical distribution. This validates Theorem 1

The accuracy of Fast-WDOPH is inferior to that of the other two unbiased schemes and sometimes its performance is poor. This is because the weighted to unweighted reduction is biased and approximate. The bias of this reduction depends on the vector pairs under consideration, which can be unpredictable.

## 4.2 Speed Comparisons

We compute the average time (in milliseconds) taken by the competing algorithms to compute 500 hashes of a given data vector for all the three datasets. Our experiments were coded in C# on Intel Xenon CPU with 256 GB RAM. Table 2 summarises the comparison. We do not include the data loading cost in these numbers and assume that the data is in the memory for all the three methodologies.

We can clearly see tremendous speedup over Ioffe's scheme. For Hist dataset with mere 768 non-zeros, our scheme is 100 times faster than Ioffe's scheme and around 5 times faster than Fast-WDOPH approximation. While on caltech101 and Oxford datasets, which are high dimensional and dense datasets, our scheme can be

|  | Hist | Caltech101 | Oxford |
|---|---|---|---|
| Mean Values | 11.94 | 52.88 | 9.13 |
| Hash Range | [1,107] | [1,487] | [1,69] |
| Bits Needed | 7 | 9 | 7 |

Table 3: The range of the observed hash values, using the proposed scheme, along with the maximum bits needed per hash value. The mean hash values agrees with Theorem 2

1500x to 60000x faster than Ioffe's scheme, while it is around 5 to 100x times faster than Fast-WDOPH scheme. Dense datasets like Caltech101 and Oxford represent more realistic scenarios. These features are taken from real applications [6] and such level of sparsity and dimensionality are more common in practice.

The results are not surprising because Ioffe's scheme is very slow $O(dk)$. Moreover, the constant are inside bigO is also large, because of complex transformations. Therefore, for datasets with high values of $d$ (non-zeros) this scheme is very slow. Similar phenomena were observed in [13], that decreasing the non-zeros by ignoring non-frequent dimensions can be around 150 times faster. However, ignoring dimension looses accuracy.

## 4.3 Memory Comparisons

Table 3 summarizes the range of the hash values and the maximum number of bits needed to encode these hash values without any bias. We can clearly see that the hash values, even for such high-dimensional datasets, only require 7-9 bits. This is a huge saving compared to existing hashing schemes which requires (32-64) bits [16]. Thus, our method leads to around 5-6 times savings in space. The mean values observed (Table 3) validate the formula in Theorem 2.

## 5 Discussions

Theorem 2 shows that the quantity $s_x = \frac{\sum_{i=1}^{D} x_i}{\sum_{i=1}^{D} m_i}$ determines the runtime. If $s_x$ is very very small then, although the running time is constant (independent of $d$ or $D$), it can still make the algorithm unnecessarily slow. Note that for the algorithm to work we choose $M_i$ to be the largest integer greater than the maximum possible value of co-ordinate $i$ in the given dataset. If this integer gap is big then we unnecessarily increase the running time. Ideally, the best running time is obtained when the maximum value, is itself an integer, or is very close to its ceiling value. If all the values are integers, scaling up does not matter, as it does not change $s_x$, but scaling down can make $s_x$ worse. Ideally we should scale, such that, $\alpha = \arg\max_\alpha \frac{\sum_{i=1}^{D} \alpha x_i}{\sum_{i=1}^{D} \lceil \alpha m_i \rceil}$ is maximized, where $m_i$ is the maximum value of co-ordinate $i$.

### 5.1 Very Sparse Datasets

For very sparse datasets, the information is more or less in the sparsity pattern rather than in the magnitude [28]. Binarization of very sparse dataset is a common practice and densified one permutation hashing [26, 27] provably solves the problem in $O(d + k)$. Nevertheless, for applications when the data is extremely sparse, and the magnitude of component seems crucial, binary approximations followed by densified one permutation hashing (Fast-DOPH) should be the preferred method. Ioeffe's scheme is preferable, dues to its exactness nature, when number the number of non-zeros is of the order of $k$.

## 6 Acknowledgements

This work is supported by Rice Faculty Initiative Award 2016-17. We would like to thank anonymous reviewers, Don Macmillen, and Ryan Moulton for feedbacks on the presentation of the paper.

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
