[Supplementary Material]

# Supplementary Material: Simple and Efficient Weighted Minwise Hashing

**Anshumali Shrivastava**
Department of Computer Science
Rice University
Houston, TX, 77005
anshumali@rice.edu

## 1 Proof of Correctness

First note that, every number between $[0, M]$ is random and equally likely in a random sampling. Therefore, for a given point $x$, at the time we stop we sample uniformly from the green region $x_{green} = \cup_{i=1}^{D}[M_i, M_i + x_i]$. Consider the index $j$ defined as,

$$j = \min\{h(x), h(y)\} \tag{1}$$

For any pair of points $x$ and $y$, consider the following three events: 1) $h(x) = h(y) = j$, 2) $h(x) > h(y) = j$ and 3) $j = h(x) < h(y)$. Observe that,

$$h(x) = h(y) = j \quad \text{if and only if} \quad r_j \in x_{green} \cap y_{green} \tag{2}$$

$$h(x) > h(y) = j \quad \text{if and only if} \quad r_j \in y_{green} - x_{green} \tag{3}$$

$$h(y) > h(x) = j \quad \text{if and only if} \quad r_j \in x_{green} - y_{green} \tag{4}$$

Since $r_j$ is uniformly chosen, we have,

$$Pr\Big(h(x) = h(y)\Big)$$

$$= \frac{|x_{green} \cap y_{green}|}{|(x_{green} \cap y_{green}) \cup (x_{green} - y_{green}) \cup (y_{green} - x_{green})|}$$

$$= \frac{|x_{green} \cap y_{green}|}{|x_{green} \cup y_{green}|} \tag{5}$$

The proof follows from substituting the values of $|x_{green} \cap y_{green}|$ and $|x_{green} \cup y_{green}|$ given by:

$$|x_{green} \cap y_{green}| = |\cup_{i=1}^{D} [M_i, M_i + x_i] \cap \cup_{i=1}^{D}[M_i, M_i + y_i]|$$

$$= |\cup_{i=1}^{D} [M_i, M_i + \min\{x_i, y_i\}]| = \sum_{i=1}^{D} \min\{x_i, y_i\} \tag{6}$$

$$|x_{green} \cup y_{green}| = |\cup_{i=1}^{D} [M_i, M_i + x_i] \cup \cup_{i=1}^{D}[M_i, M_i + y_i]|$$

$$= |\cup_{i=1}^{D} [M_i, M_i + \max\{x_i, y_i\}]| = \sum_{i=1}^{D} \max\{x_i, y_i\}, \tag{7}$$

### 1.1 Proof of Theorem 2

Expectation follows immediately from the fact that the number of sampling step taken before the process stops,which is also $h(x)$ is a geometric random variable with $p = s_x$. The collision probability follows from observing that $Pr(h(x) > k) = (1 - s_x)^k \leq \delta$ which implies $k \leq \frac{\log \delta}{\log (1-s_x)}$ yielding the required bound.

### 1.2 Proof of Corollary 1

**Proof:** The proof follows from Jensens Inequality, $\mathbb{E}(\log x) \leq \log \mathbb{E}(x)$ and second order Taylor series approximation of $\mathbb{E}(\log x) \approx \log \mathbb{E}(x) - \frac{Var(x)}{2 \log \mathbb{E}(x)^2}$