[Reviews · NeurIPS 2016]

Reviewer 1

Summary

The paper propose a rejection sampling scheme to construct weighted minwise hash. Furthermore, the paper proves the correctness (the requirement of the LSH for the L_1 metric) of the method, analyzes its advantages, and points out settings where the method is expected to work well. The experiments show significant advantages over the baselines.

Qualitative Assessment

The idea of this paper is novel, which uses reject sampling to construct a hash map. The paper is well written and easy to follow. (1) The assumption that 1/s << d is not quite realistic for real image datasets. If the dimensionality is high, the feature vector is often sparse. Otherwise, it typically means that the feature vector has a lot of redundancy. In particular, I don’t understand why the dimensionality of the features in Caltech 101 and Oxford is so large (half a million dimensions, more than the number of raw pixels). Popular features for images have only thousands of dimensions (e.g. AlexNet features). I suggest that the authors consider a more realistic setup that has an intrinsic high dimensionality. (2)It is claimed that “the proposed scheme requires much fewer bits (usually 5-9) bits”. This is only shown empirically. I wonder if there exists a sparse vector that has many 0s or small values. For this vector, the hash value will be very large (at the extreme, if x is all zeros, the sampling never stops). So the claim depends on the data distribution. Furthermore, I don’t think using fewer bits is an advantage. Fewer bits means few buckets and more examples in each bucket, which seems a drawback for retrieval operations. (3) The proposed method needs to compute m_i, which seems expensive. It takes O(nd), where n is the number of examples. In most cases, n is much larger than d. Although it is executed off-line, the cost is not negligible. I think the authors should provide details on how many examples are needed to compute (estimate) m_i for each dataset.

Confidence in this Review

2-Confident (read it all; understood it all reasonably well)


Reviewer 2

Summary

The authors propose a new algorithm of weighted minwise hashing.

Qualitative Assessment

This paper is a simple modification of the already existing methods that is (slightly) superior over existing approaches only in specific corner cases. The theoretical contribution is trivial. This paper is way below standards of any good ml conference, not just NIPS.

Confidence in this Review

3-Expert (read the paper in detail, know the area, quite certain of my opinion)


Reviewer 3

Summary

In this paper, the authors provide a new scheme for weighted minwise hashing (WMH) that can produce k hashes in O(d+k), with 'd' being the vector dimensionality. The method relies on computing maximal values per each dimension of a dataset's vectors. These values are used to construct a range of values for use in conjunction with a random sampler during the hashing function. To hash a vector, the method annotates regions within the range as 'green' using values from each dimension of the vector. It then proceeds to generate a random real sample between the minimum and maximum values of the range, and determines if that value belongs to a 'green' region. The min-hash value is the number of trials needed before a green region is found. The paper proves the correctness mathematically. It also theoretically proves the correctness, expected number of steps and memory required. They also experimentally validate the accuracy, and compare speed and bits needed with the Ioffe's scheme and fast WDOPH.

Qualitative Assessment

The paper is well written, and the presented method is sound. The improvement over state-of-the-art algorithms is significant which is demonstrated in theory and also validated by the experiments. Further, the problem tackled in the paper is an important one and may have a significant impact on indexing and search technologies. Some issues that I would like to see addressed: A technical issue in the proof: - In the proof in the supplementary material, strictly speaking (2)(3)(4) are not quite correct because the event has to consider those steps before j; Eq(5) is correct, so it is necessary to justify how from (2)(3)(4) one can deduct (5). Typos and grammatical issues. For instance, - In Line 295, Table 1 instead of Table 4 - Line 338 is grammatically wrong.

Confidence in this Review

2-Confident (read it all; understood it all reasonably well)


Reviewer 4

Summary

The paper proposes a novel weighted minwise hashing method. The idea is interesting and the proposed method is technically sound, the improved results are convincing.

Qualitative Assessment

I carefully read the reference papers, in my opinion the paper prove the proposed method is effective and useful.

Confidence in this Review

1-Less confident (might not have understood significant parts)


Reviewer 5

Summary

The paper comes up with a novel method to find the Jacaard similarity by viewing elements of vectors x,y as probabilities instead of positive integers, and uses a simple (if almost obvious in retrospect) rejection sampling / hashing method to derive these probabilities. Experimental figures are shown, and discussion about the hashing method is given.

Qualitative Assessment

The authors achieve two things in their paper: One, improving weighted minwise hashing to compute the Jacaard similarity in time an order of magnitude less than Ioffe's method. Two, which I feel is more important, a simple yet novel method which is easily understandable and implementable. To give a comparison, Ioffe's paper took a few days to read and absorb (which is understandable, since was then "the best" method so far), and one could imagine a new paper having more complex arguments / algorithm to beat Ioffe's results (in terms of accuracy and running time). This paper was remarkably easy to read and understand, and is obvious in retrospect. Thus, this paper gets high scores in technical quality, novelty, and impact. Experimental results are well presented. Some small details: Surprisingly, I found the proof in the supplementary material for Theorem 1 harder to understand (notation is a bit confusing); an easier argument would be to argue that for the number of steps (hash function output) for two pairwise vectors to be the same ; this is simply adding up the probability that : the hashes are both zero, both one, both two , both three, both four, ..... This is an infinite geometric sum, but it factors out nicely into the result required using some high school algebra. One note: seeds may be different across different languages / version of programming languages - so storing these seeds would not necessarily work all the time - although it would be easy to re-generate these hashes. Furthermore, Algorithm 3 and 4, Figure 1 are a bit dense / confusing. For example, the red green map in Figure 1 could have dotted lines extending down for each M_is for easier intuition, and better spacing of x_is, y_is. This is not to say the algorithms / figures should not be given. Rather, reading the paper gave the general idea, and it was writing out sample code / simulations in Matlab that understood what Algorithm 3 / Algorithm 4 does. Perhaps put the steps of these algorithms on the last page or in the supplementary material since the steps make the method seem much harder than it really is.

Confidence in this Review

3-Expert (read the paper in detail, know the area, quite certain of my opinion)


Reviewer 6

Summary

none

Qualitative Assessment

This paper proposed a simple hash scheme, which is proved to satisfy the Weighted Minwise Hashing property. The method simply checked whether a random value is within some certain range and recorded the number of repeated steps as hash value. The author then gives a proof that the method is exact. It is theoretically sound and efficient. Experiments show that the proposed method achieves almost the same performance as the state-of-the-art methodology, while being much faster. Both the theoretic and empirical parts of this paper look sound and convincing. Overall, I recommend poster acceptance. I would only have a minor remark: -There are some works of learning to hash on vision tasks, e.g. spectral hashing, angular quantization. Could you please mention the relations to these methods? It would be better to have an overview for researchers who are more specialized in that domain.

Confidence in this Review

2-Confident (read it all; understood it all reasonably well)